# The Critical Role of Host and Bacterial Extracellular Vesicles in Endometriosis

**DOI:** 10.3390/biomedicines12112585

**Published:** 2024-11-12

**Authors:** Michaela Wagner, Chloe Hicks, Emad El-Omar, Valery Combes, Fatima El-Assaad

**Affiliations:** 1Microbiome Research Centre, St George and Sutherland Clinical School, UNSW Sydney, Sydney, NSW 2052, Australia; michaeladwagner99@gmail.com (M.W.); chloe.hicks@adelaide.edu.au (C.H.); e.el-omar@unsw.edu.au (E.E.-O.); 2Malaria and Microvesicles Research Group, School of Life Science, Faculty of Science, University of Technology Sydney, Ultimo, Sydney, NSW 2007, Australia; valery.combes@uts.edu.au

**Keywords:** endometriosis, extracellular vesicles, host extracellular vesicles, bacterial extracellular vesicles, microbiome, microbiota, inflammation, immunity, biomarkers

## Abstract

Endometriosis is a chronic, inflammatory, oestrogen-dependent disorder that is defined by the presence of endometrium-like tissue in the extra-uterine environment. It is estimated to affect approximately 10% of women of reproductive age, and the cause is still largely unknown. The heterogenous nature and complex pathophysiology of the disease results in diagnostic and therapeutic challenges. This review examines the emerging role of host extracellular vesicles (EVs) in endometriosis development and progression, with a particular focus on bacterial extracellular vesicles (BEVs). EVs are nano-sized membrane-bound particles that can transport bioactive molecules such as nucleic acids, proteins, and lipids, and therefore play an essential role in intercellular communication. Due to their unique cargo composition, EVs can play a dual role, both in the disease pathogenesis and as biomarkers. Both host and bacterial EVs (HEVs and BEVs) have been implicated in endometriosis, by modulating inflammatory responses, angiogenesis, tissue remodelling, and cellular proliferation within the peritoneal microenvironment. Understanding the intricate mechanisms underlying EVs in endometriosis pathophysiology and modulation of the lesion microenvironment may lead to novel diagnostic tools and therapeutic targets. Future research should focus on uncovering the specific cargo, the inter-kingdom cell-to-cell interactions, and the anti-inflammatory and anti-microbial mechanisms of both HEVs and BEVs in endometriosis in the hope of discovering translational findings that could improve the diagnosis and treatment of the disease.

## Lay Summary

Endometriosis is a chronic condition where uterus-like tissue grows outside the uterus, causing inflammation and pain. It affects about 10% of women of reproductive age, but its exact cause is still unclear. The disease is complex, with varying presentations, making it hard to diagnose and treat. Tiny host cell-derived particles called extracellular vesicles (HEVs) are involved in the communication between cells and play a role in endometriosis by affecting inflammation, blood vessel growth, tissue changes, and cell growth. Bacterial extracellular vesicles (BEVs) are tiny particles released by bacteria that are emerging as key players in inflammatory conditions. Very little is known about their role in endometriosis. However, studying these EVs more closely could lead to better ways to diagnose and treat endometriosis.

## 1. Introduction

Endometriosis is a chronic, inflammatory, and oestrogen-dependent condition that is defined by the presence of endometrium-like tissue outside of the uterine cavity [1]. It is estimated to affect 10% of women of reproductive age and is characterised by painful debilitating symptoms and infertility but can also present asymptomatically [2]. The heterogenous nature of the disease, combined with the lack of non-invasive diagnostic tools, results in a diagnostic delay on average of 7 years [3]. Endometriosis is often staged using the American Society for Reproductive Medicine (ASRM) classification system, where the lowest score, stage 1, corresponds to minimal lesions found and the highest score, stage 4, corresponds to widespread severe lesions and adhesions, often impacting fertility and bowels [4]. This system helps describe the extent and severity of the disease, although the stages do not necessarily correlate with symptoms and the impact on the quality of life. There is currently no known cure, and treatment options such as laparoscopic excision of lesions and hormonal treatments have limited efficacy and undesirable side effects and are not guaranteed to prevent symptom recurrence [2]. Ultrasound or MRI imaging is largely limited to deep infiltrating endometriosis, and a negative scan does not exclude superficial endometriosis [5]. 

There are several theories to describe the pathophysiology of endometriosis, such as retrograde menstruation, genetics, immune dysregulation, hormonal imbalances, and coelomic metaplasia, yet the cause of the disease remains poorly understood [6]. The study of extracellular vesicles in endometriosis is an emerging area of research that has the potential to improve the understanding of endometriosis pathophysiology, while also providing new avenues for diagnostics and therapeutics [7]. 

## 2. Extracellular Vesicles 

Extracellular vesicles (EVs) are nano-sized, 20–1000 nm, bilipid enclosed membrane-bound structures that are released by all cell types, of both eucaryotes and procaryotes, into their extracellular environment [8,9]. They can carry cargo, which is reflective of the phenotype of their origin cell at the time of vesiculation [10,11]. EVs can travel to close or distant sites around the body and regulate cell function [12]. Due to this trait, their role in intercellular communication, immune response, disease pathogenesis, novel biomarkers, and drug delivery is increasingly being explored [10,13,14]. EVs play an important role in intercellular communication as they can effectively transport active biomolecules such as nucleic acids, proteins, amino acids, and metabolites to target distant or close cells and induce various physiological responses [15,16,17]. As such, EV can impact and regulate physiological processes such as immune responses and angiogenesis [16,17]. In this review, we refer to human host EVs (HEVs, eucaryotic-origin) and bacterial EVs (BEVs, procaryotic-origin), and we summarise their distinct characteristics, functions (Table 1), and implications for disease and therapeutic applications in endometriosis and in selected inflammatory conditions. 

## 3. Endometriosis Pathogenesis 

While the cause of endometriosis remains unknown, there are several theories that have been proposed to explain the pathophysiology of this disease. Such theories include the coelomic metaplasia theory, the embryonic rest theory, and the theory of retrograde menstruation [25]. The theory of retrograde menstruation, initially proposed by Sampson in the 1920s, suggests that there is a backward flow of menstrual fluid and endometrial tissue via the fallopian tubes into the peritoneal cavity [26]. This theory is attractive due to the common anatomical distribution of lesions, and the higher prevalence of endometriosis in women with obstructed menstrual outflow tracts [27]. While this theory explains the physical presence of endometrial tissue in the peritoneal cavity, it does not elucidate how lesions are established and develop [27]. 

The microbiota is the collection of microorganisms living on and within the human host, contributing to a range of pathophysiological functions implicated in our health [28,29]. There is emerging evidence that supports the role of microbiota in the development and progression of endometriosis [30,31,32]. Firstly, the gut microbiota plays an essential role in chronic disease by regulating immunity and inflammation in the human body [33]. Numerous studies have demonstrated the gut microbiota is altered in endometriosis, and that dysbiosis of the microbiome can result in disrupted immune responses, oestrogen and hormonal imbalances, and cause chronic inflammation [31,32,34]. It is therefore suggested that the gut microbiota may impact endometriosis pathophysiology by contributing to impaired clearance of endometrial fragments, adhesion, invasion, and angiogenesis [34]. It has been demonstrated in mouse models of endometriosis that certain gut microbiota and microbiota-derived metabolites promote the growth of lesions, while antibiotic therapy reduces lesion size as well as inflammatory markers in the peritoneal fluid [35]. Additionally, some gut microbiota-derived short-chain fatty acids have been found to be protective against endometriosis progression in mice [36]. The estrobolome is defined as the community of microbes with genes encoding oestrogen-metabolising enzymes and can therefore increase circulating estrogens [37,38]. It is suspected that the estrobolome is involved in endometriosis due to the oestrogen-dependent nature of the condition [39]. To date, one study has found that faecal samples of patients with endometriosis had higher levels of oestrogen; however, the reason for this finding is unclear [40].

There is also evidence of the involvement of the urogenital microbiota in endometriosis pathogenesis. Firstly, the ‘bacterial contamination hypothesis’, proposed by Khan et al. in 2016, suggests that the growth of endometriosis lesions is mediated by the LPS/Toll-like receptor 4 (TLR4) cascade due to high levels of *Escherichia coli* and therefore endotoxins in the menstrual efflux in peritoneal fluid [41]. More recently, it was demonstrated that *Fusobacterium* infection of the endometrium has a pathogenic role in the development of ovarian endometriosis [42]. Briefly, it was shown that infection of endometrial cells led to the activation of transforming growth factor-β (TGF-β) signalling, causing quiescent fibroblasts to transform into transgelin (TAGLN)-positive myofibroblasts, giving the cells the ability to adhere and proliferate in vitro [42]. One flaw of the bacterial contamination hypothesis is that the vaginal microbiome is largely dominated by *Lactobacillus*, which is believed to be protective and prevent invasion of pathogens [43].

## 4. Endometriosis and Host Extracellular Vesicles (HEVs)

There are three main classifications of HEVs based on their biogenesis, origin, size, function, and content: exosomes, microvesicles, and apoptotic bodies [44] (Table 1).

HEVs appear as critical mediators in the pathogenesis of endometriosis [45]. They are involved in cell communication [46], immune modulation though suppression of natural killer (NK) cell activity [47] and modulation of macrophage activity [48], and prevention of endometrial lesion clearance by the immune system [47] and promotion of inflammation and tissue adhesion via proinflammatory cytokine transport. The composition of the cargo identified from various studies suggest that EVs also play a role in cell proliferation, migration, fibrosis, and angiogenesis [7,49]. Collectively, these activities promote the survival and growth of ectopic endometrial tissue and the lesions in endometriosis (Figure 1).

There are differences in EV cargo within the endometrium, plasma, serum, and peritoneal fluid of patients with and without endometriosis [7]. EVs isolated from various biological specimen, including serum, endometrium, and peritoneal fluid, all contained microRNA, small, non-coding RNA molecules, and cargo related to angiogenesis, cell proliferation, cell migration, and immunomodulation [7]. In some cases, the levels of serum exosomes correlated with disease severity within the endometriosis group [50]. Vagina-derived EVs impact human sperm function and fertility [51]. These vagina-derived EVs contribute to an imbalance between Th17 cells (promote inflammation) and regulatory T cells (help control immune responses), potentially worsening inflammation and immune dysfunction in endometriosiss [51].

Few studies have investigated the cargo of EVs for potential biomarkers of endometriosis [52,53,54]. Vascular endometrial growth factor (VEGF-C) is one potential biomarker and was found to be upregulated in the endometrial cells, serum, and peritoneal fluid of patients with endometriosis [54,55]. VEGF-C was transported by EVs and contributed to increased lymphangiogenesis, when the lymphatic vessel density is increased within the endometrium, allowing the lesions to travel ectopically [56].

There is also limited research on the potential therapeutic role of EVs in endometriosis. Within mouse models, nanovesicles (artificially produced EVs) derived from macrophages were shown to inhibit the development of endometriosis [57]. Additionally, microRNA-214 has been shown to play an important role in the fibrotic disease and its expression is decreased within ectopic endometrial stromal cells. Injecting microRNA-214 enriched exosomes from endometrial stromal cells into an endometrial mouse model showed a decrease in the expression of fibrosis-associated proteins [58]. Although preliminary, this research paves way for emerging therapeutic roles for EVs within endometriosis. Despite being in its infancy, the role of EVs as potential diagnostic and therapeutic tools shows promising outcomes for those affected by endometriosis.

## 5. Bacterial Extracellular Vesicles

EVs derived from bacteria and filled with bioactive bacterial components from the parent bacteria are referred to bacterial extracellular vesicles (BEVs). BEVs often range from 20 to 400 nm in diameter and can be categorised as outer-membrane proteins or membrane vesicles depending on whether they originate from Gram-negative or -positive bacteria, respectively [59] (Table 2). BEVs not only play an important role in the bacteria-to-bacteria crosstalk, but also in host-to-bacteria interactions, delivering effector molecules that modulate pathways. Depending on their origin, BEVs can phenotypically influence the host both positively and negatively. For example, in a mouse model of inflammatory bowel disease, *Akkermansia muciniphila*-derived BEVs were shown to inhibit the production of interleukin 6, reducing inflammation within the host and preventing colitis [60]. This bacterium is typically seen as a beneficial bacterium, and hence, its BEVs have a positive effect on the host [61]. Alternatively, BEVs from *Acinetobacter baumanni*, a bacterium typically associated with disease, have been shown to promote inflammation in mice through the activation of Toll-like receptors, which triggers an inflammatory cascade within the pulmonary system [62,63].

A dysbiotic gut refers to an imbalance of the microbiome to favour bacteria which promote disease and inflammation [64]. A dysbiotic gut occurs in many diseases such as obesity, inflammatory bowel disease, and rheumatoid arthritis and alters the gut lumen creating a ‘leaky gut’ that allows EVs and BEVs to penetrate and enter the circulatory system [64] (Figure 2). Here, they can travel to various sites across the body and phenotypically influence the host, depending on their cargo [65] (Figure 2).

**Table 2 biomedicines-12-02585-t002:** Potential mechanisms by which bacterial extracellular vesicles (BEVs) regulate cytokine production in endometriosis.

Feature	Role in Endometriosis	Potential Mechanism of BEVs
**↑** **IL-17**	Elevated in peritoneal fluid and regulates macrophage recruitment Drives macrophage polarisation towards the M2 phenotype [48]	*E. coli*-derived EVs isolated from indoor dust induce chronic obstructive pulmonary disease in mice via neutrophilic inflammation mediated by IL-17A [66]
**↑** **TNF**	Elevated in serum [67] Elevated in peritoneal fluid [68]	Airway exposure to *E. coli* EV increased the production of proinflammatory cytokines, such as TNF and IL-6 [66] OMVs released by *Aggregatibacter actinomycetemcomitan*s contain exRNAs that promote TNF [69] fMVs evoke the release of TNF by THP-1 cells in a dose-dependent matter. Also, a significant positive correlation was found between Actinobacteria/γ-Proteobacteria-derived vesicles and the release of TNF [70]
**↑** **IL-6**	Elevated in peritoneal fluid and serum [68,71]	Airway exposure to *E. coli* EV increased the production of proinflammatory cytokines, such as TNF-α and IL-6 [66] OMVs derived from pathogenic *E. coli* induces elevated IL-6 levels in human umbilical vein endothelial cells [72]
**↑** **IL-1** **β**	IL-1β levels in peritoneal fluid of endometriosis women were higher than in controls [73]	*P. gingivalis*, *T. denticola*, and *T. forsythia* OMVs on monocytes induced NF-κB activation and increased TNFα, IL-8, and IL-1β cytokine secretion [74]
**↑** **IL-10**	Serum level of IL-10 in patients with endometriosis was significantly higher [75]	*P. gingivalis* OMVs were also found to induce anti-inflammatory IL-10 secretion [74]

## 6. Endometriosis and Bacterial Extracellular Vesicles (BEVs)

It is difficult to determine the role of BEVs in endometriosis due to the limited studies exploring their role. The outer layer of BEVs enables them to travel over large distances without being destroyed, and it has been widely demonstrated that EVs are capable of transporting virulence factors derived from pathogenic bacteria [76]. Therefore, it may be possible that BEVs contribute to endometriosis pathogenesis by transporting virulence factors, resulting in an inflammatory pelvic environment.

To date, there is only one study that explores the influence of BEVs on endometriosis [77]. A microbiome analysis was conducted of the peritoneal fluid from women with and without ovarian endometriosis [78]. The origin of BEVs within the peritoneal fluid was analysed further and showed that the microbial composition of BEVs was significantly different between women with stage 3 to 4 endometriosis when compared to that of women without endometriosis. This highlights BEVs’ potential role in the pathogenesis and progression of endometriosis (Figure 1). However, the origin of BEVs or the function of the cargo they contained was not analysed. Therefore, it is difficult to conclude more about this relationship. It is also important to recognise that controversy exists regarding the presence of a microbiome within the peritoneal fluid. Regardless, this study highlights a novel relationship between BEVs and endometriosis as well as a gap in knowledge and a need to further explore this relationship through further clinical studies.

Another potential mechanism for BEVs to enact their function is via chemokine and cytokine stimulation (Table 2). It has been established that interleukin-8 (IL-8) expression is upregulated in endometriosis lesion tissue and is closely related to disease progression [79]. It has also been established that there is a significant increase in myeloid-derived suppressor cells (MDSCs) in the peritoneal fluid and blood of patients with endometriosis, with chemokines CXCL1, 2, and 6 acting as the key mediators for the recruitment of these cells. MDSCs enhance angiogenesis and thus contribute to disease progression [79]. While investigating atherosclerosis, it was found that *Porphyromonas gingivalis*-derived vesicles upregulated the expression of CXCL1, 2, and 8 in human umbilical vein endothelial cells, and that the vesicles were more potent in inducing an inflammatory response compared to the bacterial cells themselves. It was also found that the invasive ability of the cells and vesicles was correlated with an elevated expression of IL-8 [80]. In the context of cystic fibrosis, it has been demonstrated that IL-8 secretion by lung epithelial cells is elicited by BEVs derived from *Pseudomonas aeruginosa* [81]. While the stimulation of an inflammatory response in endometriosis has not yet been demonstrated, this may be a potential mechanism through which systemic inflammation is caused.

BEVs may also be involved in endometriosis pathogenesis via macrophage polarisation. Previous studies have reported that the peritoneal microenvironment in endometriosis is characterised by a M2 tissue repair predominant macrophage phenotype, which enables immune evasion and prevents clearance of endometrial tissue from the peritoneal cavity, thus allowing development of lesions [48,82]. A 2022 study by Liang et al. found that *Clostridium butyricum*-derived EVs could regulate macrophage polarisation towards the M2 phenotype in a murine model of colitis [83]. Investigating this concept in the context of endometriosis could provide valuable insights into the mechanisms of endometriosis pathogenesis.

## 7. Extracellular Vesicles and Bacterial Extracellular Vesicles in Other Conditions

### 7.1. Gynaecology

The role of EVs within gynaecological conditions such as polycystic ovarian syndrome, preeclampsia, and adenomyosis has been explored; however, limited research exists on the role of BEVs within these gynaecological conditions [84,85,86,87]. To our knowledge, one study explores the relationship between BEVs and gynaecological conditions. *Akkermansia muciniphila* was shown to be decreased in patients with preeclampsia. When *Akkermansia muciniphila* BEVs were transferred to mice’s gastrointestinal tract, they migrated to the placenta and decreased preeclamptic symptoms [88]. Additionally, individuals with periodontitis are more inclined to suffer from preeclampsia, and this is linked to the increased presence of BEVs that travel to remote sites to promote inflammation [89].

### 7.2. Inflammation

Growing evidence suggests a relationship between BEVs and inflammation within the host. Recently, an in vivo study demonstrated that BEV derived from *Escherichia coli* (*E. coli*) induced the host release of EVs carrying lipopolysaccharides (LPSs), i.e., large molecules from outside of *E. coli*, which enhanced the inflammatory response through Toll-like-receptor 4 (TLR) signalling [90,91,92]. A dysbiotic microbiome will often favour more pathogenic bacteria, and this in turn will produce BEVs that will likely promote inflammation. BEVs have been shown to play an important role in regulating intestinal homeostasis, and their role in inflammatory bowel disease pathogenesis is being uncovered [93]. Similarly, *Escherichia coli* BEVs were shown to upregulate TLR signalling and induce inflammation [94].

### 7.3. Cancer

EVs’ ability to package cargo and travel to distant sites allows them to play an important role in various cancers [84]. EVs secreted by cancer cells contain functional oncoproteins and oncogenic RNA that work to mimic the primary tumour traits [95]. There is increasing evidence suggesting that the microbiome plays a strong role in cancer evolution and immunosurveillance. It is theorised that BEVs could play a role in tumour promotion through invoking tolerogenic immune reprogramming of the tumour’s microenvironment [96]. Multiple studies have shown that BEVs will accumulate within the tumour’s microenvironment, suggesting a potential role in cancer progression [97].

## 8. Conclusions and Future Directions

EVs and BEVs engage in a complex interplay between the microbiome and the immune system to potentially facilitate the progression of endometriosis and modulate the lesion microenvironment. EVs can influence both local and systemic environments, while BEVs promote inflammation and facilitate the growth of endometriosis lesions through protective mechanisms. EVs and BEVs can be collected and analysed from various biological samples and may offer a snapshot of the disease stage, which can be particularly powerful for the development of early diagnostic tools [59]. Their anti-microbial activity and anti-inflammatory effects can be potentially synergised to dampen the impact of endometriosis. Further research is warranted to explore the precise mechanisms by which BEVs and EVs contribute to the onset and/or progression of endometriosis. In addition, longitudinal studies characterising the dynamic release of these vesicles over the course of their cyclical flare ups would ascertain their biomarker potential in endometriosis and help with the development of scalable production for targeted treatments and improve outcomes for patients, particularly those with inflammation-related pain.

## Figures and Tables

**Figure 1 biomedicines-12-02585-f001:**
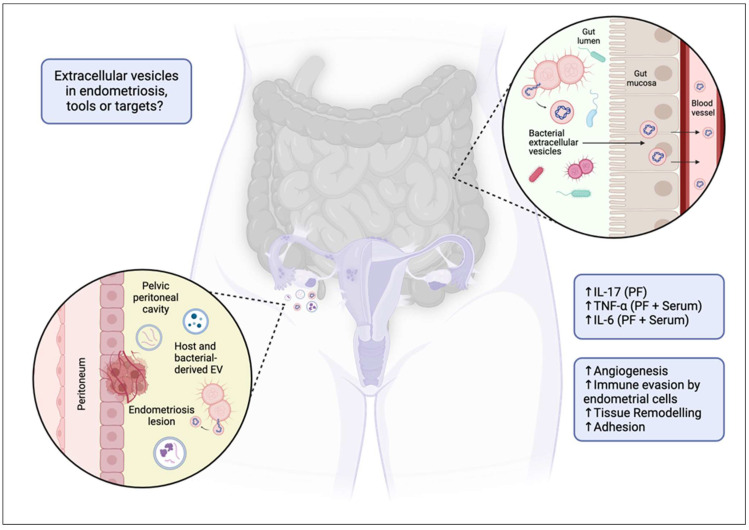
Extracellular vesicles as targets or tools in endometriosis? Proposed mechanism of extracellular vesicle (EV) and bacterial EV (BEV) involvement in endometriosis. The diagram illustrates how a dysbiotic gut may facilitate the translocation of BEVs across the gut mucosa into systemic circulation. These vesicles, along with host EVs, can migrate to the pelvic peritoneal cavity, contributing to the pathophysiology of endometriosis by promoting angiogenesis, immune evasion by endometrial cells, tissue remodelling, and adhesion. The associated increase in inflammatory markers (IL-17, TNF-α, IL-6, etc.) in the peritoneal fluid (PF) and serum is also indicated, highlighting their potential role in disease progression. ↑ = increased.

**Figure 2 biomedicines-12-02585-f002:**
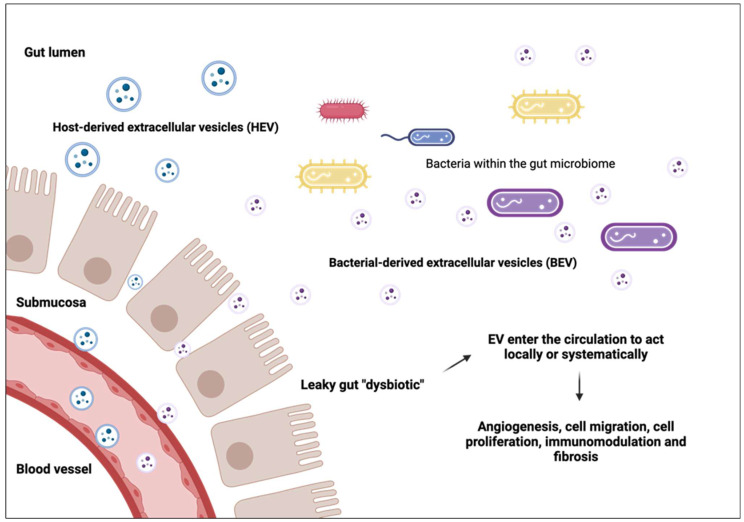
Extracellular vesicles play a role in various pathophysiological functions. A leaky ‘dysbiotic’ gut allows extracellular vesicles (EVs) and bacterial extracellular vesicles (BEVs) to enter the circulation to act locally or systematically.

**Table 1 biomedicines-12-02585-t001:** Summary of the characteristics of host extracellular vesicles (HEVs), Gram-negative bacterial extracellular vesicles (BEVs), and Gram-positive BEVs.

Characteristic	HEVs	Gram-Negative BEVs	Gram-Positive BEVs
**Types**	Exosomes Microvesicles (MVs) Apoptotic bodies	Outer-membrane vesicles (OMVs) Outer–inner-membrane vesicles (O-IMVs) Vesicle chains	Membrane vesicles (MVs) Tube-shaped vesicles (TSVs) Cytonemes/nanotubes
**Size range**	Exosomes: 30–150 nm Microvesicles: 100–1000 nm Apoptotic bodies: 500–2000 nm	OMVs: 20–300 nm O-IMVs: varies Vesicle chains: varies	MVs: 20–250 nm TSVs: varies Cytonemes/nanotubes: varies
**Origin**	Exosomes: derived from endosomal membranes Microvesicles: bud from the plasma membrane Apoptotic bodies: formed during apoptosis	OMV: outer membrane of Gram-negative bacteria O-IMV: both outer and inner membranes Vesicle chains: outer membrane of Gram-negative bacteria	MVs: cytoplasmic membrane, must pass through the thick peptidoglycan layer TSVs: formed from the bacterial membrane under specific conditions Cytonemes/fanotubes: membrane extensions
**Membrane Composition**	Exosomes: phospholipids, cholesterol, and membrane proteins Microvesicles: plasma membrane lipids and proteins Apoptotic bodies: cellular membrane remnants	OMVs: lipopolysaccharides (LPS), outer-membrane proteins, phospholipids, and peptidoglycan fragments O-IMVs: components from both outer and inner membranes Vesicle chains: outer membrane proteins and lipids	MVs: lipoteichoic acids, peptidoglycan, membrane proteins, and cytoplasmic membrane lipids TSVs: similar to MVs but with tubular structure Cytonemes/nanotubes: membrane lipids and associated proteins
**Contents**	Exosomes: proteins, lipids, RNA, DNA, and microRNAs Microvesicles: plasma membrane proteins, lipids, and cytoplasmic content Apoptotic bodies: cellular debris, DNA, and organelles	OMV: LPS, outer membrane proteins, periplasmic proteins, DNA, RNA, and virulence factors O-IMVs: mixed content including cytoplasmic material Vesicle chains: virulence factors, enzymes, DNA, and RNA	MV: cytoplasmic membrane components, enzymes, toxins, DNA, and RNA TSVs: DNA, proteins, and other molecules Cytonemes/nanotubes: various proteins and signalling molecules
**References**	[18,19,20]	[21,22,23]	[21,22,24]

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
