# Peer review of "The Critical Role of Host and Bacterial Extracellular Vesicles in Endometriosis"

_biomedicines, 2024, doi:10.3390/biomedicines12112585_

Round 1
Reviewer 1 Report
Comments and Suggestions for Authors
In this review, the authors discuss the emerging role of host-derived extracellular vesicles in the development and progression of endometriosis, with a particular focus on bacterial-derived extracellular vesicles. The subject of this review is suitable for the “Biomedicines” journal. The authors argue that focus should be placed on uncovering the specific cargo, interactions, and mechanisms of both host-derived extracellular vesicles and bacterial-derived extracellular vesicles to discover translational findings that could improve the diagnosis and treatment of endometriosis. The review is well-designed and presented, but I offer a few corrections to increase the scientific value of the manuscript. After the authors address these corrections, the manuscript can be accepted.
- Table 1 should be cited from relevant sources.
- Only a few studies of the authors have been cited (4 studies). If there are more previous studies of authors, they should be cited.
- Information under the title "Extracellular vesicles and bacterial-extracellular vesicles in other conditions" should be presented as subheadings. For example,
7. Extracellular vesicles and.......
7.1. Gynaecology
7.2. Inflammation
7.3. Cancer
- The review has been well presented, but the conclusion section should be improved, especially For early diagnosis of endometriosis.
Also;
-The topic and references are appropriate.
-Tables and figures are presented clearly and understandably.
Reviewer 2 Report
Comments and Suggestions for Authors
The manuscript titled “The critical role of host and bacterial-derived extracellular vesi- 2 cles in endometriosis” was done by Michaela Wagner, it is interesting and important for the patients with endometriosis, but the review was not organized and need more revisions about comprehensive summarize related literature. In addition, same minor revisions needed to be concerns, such as delete excessive space between two words, added a space between the number and unit. In addition, several statement are not correct, such as In line 183, “EV derived from the microbiota are referred to bacterial extracellular vesicles (BEV).” These concerns should be confirmed and revised.
